# Therapeutic Role of Tocilizumab in SARS-CoV-2-Induced Cytokine Storm: Rationale and Current Evidence

**DOI:** 10.3390/ijms22063059

**Published:** 2021-03-17

**Authors:** Corrado Pelaia, Cecilia Calabrese, Eugenio Garofalo, Andrea Bruni, Alessandro Vatrella, Girolamo Pelaia

**Affiliations:** 1Department of Health Sciences, University “Magna Graecia” of Catanzaro, 88100 Catanzaro, Italy; pelaia.corrado@gmail.com; 2Department of Translational Medical Sciences, University of Campania “Luigi Vanvitelli”, 80131 Naples, Italy; cecilia.calabrese@unicampania.it; 3Department of Experimental and Clinical Medicine, University “Magna Graecia” of Catanzaro, 88100 Catanzaro, Italy; eugenio.garofalo@unicz.it (E.G.); andreabruni87@gmail.com (A.B.); 4Department of Medicine, Surgery and Dentistry, University of Salerno, 84084 Salerno, Italy; avatrella@unisa.it

**Keywords:** SARS-CoV-2, ARDS, cytokine storm, IL-6, tocilizumab

## Abstract

Among patients suffering from coronavirus disease 2019 (COVID-19) syndrome, one of the worst possible scenarios is represented by the critical lung damage caused by the severe acute respiratory syndrome coronavirus-2 (SARS-CoV-2)-induced cytokine storm, responsible for a potentially very dangerous hyperinflammatory condition. Within such a context, interleukin-6 (IL-6) plays a key pathogenic role, thus being a suitable therapeutic target. Indeed, the IL-6-receptor antagonist tocilizumab, already approved for treatment of refractory rheumatoid arthritis, is often used to treat patients with severe COVID-19 symptoms and lung involvement. Therefore, the aim of this review article is to focus on the rationale of tocilizumab utilization in the SARS-CoV-2-triggered cytokine storm, as well as to discuss current evidence and future perspectives, especially with regard to ongoing trials referring to the evaluation of tocilizumab’s therapeutic effects in patients with life-threatening SARS-CoV-2 infection.

## 1. Introduction

The widespread outbreak of the infection caused by SARS-CoV-2, associated with the consequent COVID-19 syndrome, currently represents the most tremendous health emergency worldwide [1]. The phenotypic traits of this infection span from a complete absence of symptoms, through mild and moderate clinical manifestations (fever, cough, asthenia, neurological symptoms such as headache and dizziness), up to serious pneumonia possibly leading to lung failure, acute respiratory distress syndrome (ARDS), and eventually death [2]. With regard to the causes of such different clinical expressions, a crucial role is played by complex interactions between the virus and the individual features of the host immune system. Asymptomatic infected subjects, as well as patients complaining of only mild disease, mount an effective immune response mediated by both T and B lymphocytes. In particular, the development of memory T helper 1 (Th1) and T follicular helper (Tfh) cells induces B lymphocytes to differentiate into plasma cells producing high amounts of antiviral neutralizing antibodies [3,4]. In addition to CD4^+^ T cells, also CD8^+^ T cells undergo a significant expansion upon SARS-CoV-2 infection, and both CD4^+^ and CD8^+^ T lymphocytes synthesize the antiviral protein interferon-γ [2,5]. These reactive responses are mimicked by the mechanisms of action of currently available anti-SARS-CoV-2 vaccines [6].

When the human organism is not able to implement an efficient adaptive immune response capable of clearing the viral infection, the possible predominance and persistence of innate immune pathways can drive the development and amplification of a hyperinflammatory state, sustained by a massive release of cytokines and chemokines [7]. This cytokine storm often occurs in the most severe COVID-19 cases, and is associated with lymphopenia responsible for defective T and B cell-dependent immune responses, ineffective viral clearance, and destruction of inflamed tissues [3]. Such events are due to a pathologic inefficiency of both cell-mediated and antibody-operated arms of the immune system, which originates from an exhaustion of individual immunity possibly favored by several factors including aging, comorbidities, immunosuppressive therapies, and also a very high viral load [2,3]. The main target organ of the COVID-19-induced cytokine storm is the lung, possibly affected by SARS-CoV-2-led interstitial pneumonia, ARDS, and vascular endothelial damage [2,8,9].

Cytokine storm is characterized by the overexpression of multiple proinflammatory cytokines, chemokines, and growth factors such as interleukin-1β (IL-1β), interleukin-6 (IL-6), interleukin-8 (IL-8), interferon γ-induced protein 10 (IP-10), granulocyte-macrophage colony-stimulating factor (GM-CSF), tumor necrosis factor-α (TNF-α), and transforming growth factor-β (TGF-β) [8,10]. Within the complex inflammatory context of SARS-CoV-2-induced cytokine storm (Figure 1), a prominent pathogenic role is played by IL-6, whose high blood levels appear to be associated with an increased mortality risk [10]. Indeed, IL-6 is a pleiotropic cytokine which contributes to stimulating the production of the acute phase C-reactive protein (CRP), to impairing the Th1 cell-mediated anticoronavirus response, to inhibiting the physiologic actions of CD8^+^ cytotoxic cells and natural killer (NK) cells, as well as to promoting the differentiation of Th17 lymphocytes [7,11,12]. Moreover, elevated IL-6 serum concentrations seem to predispose to the development of severe lung tissue damage [13]. Therefore, IL-6 inhibition looks like an attractive therapeutic strategy in order to attenuate the dramatic consequences of COVID-19-associated cytokine storm. In this regard, the IL-6 receptor antagonist tocilizumab, a humanized monoclonal antibody already approved for treatment of rheumatoid arthritis, is currently under clinical investigation in COVID-19 patients [14,15]. However, ongoing trials are yielding mixed results, not leading to convincing and conclusive evidence [16]. In particular, though tocilizumab may improve the oxygenation pattern of hospitalized COVID-19 patients, their mortality rate does not appear to be significantly affected by this biologic drug [16]. Hence, further clinical studies should be carried out with the aim of expanding our not yet solid knowledge about the real impact of tocilizumab on the most severe consequences of SARS-CoV-2 infection.

Based on the above considerations, the present review article aims to discuss the immunopathology of the cytokine storm caused by SARS-CoV-2 infection, as well as to focus on the molecular rationale and the clinical efficacy of tocilizumab use in severe COVID-19 patients.

## 2. Cellular and Molecular Pathophysiology of SARS-CoV-2-Driven Cytokine Storm

### 2.1. Pathogenic Mechanisms Underlying SARS-CoV-2 Infection

SARS-CoV-2 is an enveloped, positive-sense, and single-stranded RNA β-coronavirus, characterized by a crown-like shape similar to coronaviruses responsible for severe acute respiratory syndrome (SARS) and Middle East respiratory syndrome (MERS) [17,18]. The phospholipid bilayer structure of the envelope stems from host cell membranes. The nucleocapsid contains the RNA genome of 27.9 kb, which encodes several nonstructural accessory proteins and four structural essential proteins including the envelope E protein, the matrix M protein, the nucleocapsid N protein, and the spike S glycoprotein [19]. The latter specifically binds to the cell membrane receptor angiotensin-converting enzyme 2 (ACE2), densely expressed by airway epithelial cells, alveolar epithelial cells, lung macrophages, and vascular endothelial cells [20,21,22]. SARS-CoV-2 attachment to ACE2 is associated with S protein cleavage operated by the host transmembrane serine protease TMPRSS2, which greatly facilitates viral–cellular interactions [17,20]. Indeed, S glycoprotein includes an S1 subunit targeting infected cells and an S2 subunit, activated upon TMPRSS2-catalyzed cleavage, which is responsible for the fusion process between SARS-CoV-2 and host cell plasma membrane [23]. Such a fusion allows virus entry inside the cytoplasm of infected cells. Once penetrated into host cells, single-stranded SARS-CoV-2 RNA is recognized by endosomal toll-like receptor 7 (TLR7) [24]. Upon virus entry, the host cell machinery is exploited by SARS-CoV-2, so that its RNA can be replicated by RNA-dependent RNA polymerase (RdRp), transcribed and translated into viral proteins [25]. The latter are then inserted within the membrane complex consisting of the endoplasmic reticulum and Golgi apparatus, where SARS-CoV-2 essential proteins are assembled and subsequently combined with genomic RNA, thus giving rise to newly formed vesicles containing mature viral particles, that fuse with host plasma membrane and are exocytosed [25].

### 2.2. Innate and Adaptive Immune Responses Triggered by SARS-CoV-2

Within the molecular context of class I major histocompatibility complex (MHC-I), SARS-CoV-2 structural proteins are presented by infected epithelial cells to CD8^+^ cytotoxic T lymphocytes, which release proapoptotic factors such as perforin and granzymes, thereby inducing the apoptosis of virus-targeted cells [3]. Moreover, subepithelial dendritic cells and tissue macrophages present SARS-CoV-2 antigens, embedded inside class II major histocompatibility complex (MHC-II), to CD4^+^ T lymphocytes, thus promoting their commitment towards Th1 and memory Tfh cells, which in turn promote the differentiation of B lymphocytes into plasma cells producing specific anti-SARS-CoV-2 IgM and IgG antibodies [3]. These neutralizing antibodies act together with antiviral type I interferons produced by both CD4^+^ and CD8^+^ T cells. In most SARS-CoV-2-infected subjects, such a synergistic cooperation between the cellular and humoral arms of the immune system provides an effective resolution of viral infection [22]. In fact, neutralized SARS-CoV-2 and apoptotic host cells are phagocytosed and cleared by alveolar macrophages, with a consequent healing associated with irrelevant lung damage [22].

However, in some cases a dysfunctional immune response can lead to severe pulmonary and also systemic injuries [2,22]. These inefficient and harmful immune mechanisms can be associated with lymphopenia and activation of aberrant innate and adaptive immunity pathways. With regard to lymphopenia, it has been suggested that in critically ill COVID-19 patients, lymphocyte depletion may occur as a result of both apoptosis and pyroptosis [3,22]. Indeed, T cell apoptosis was previously detected in some patients with SARS and MERS diseases, who were characterized by elevated plasma levels of Fas-ligand, as well as by high numbers of caspase-3 positive CD4^+^ and CD8^+^ T lymphocytes [3,26]. Lymphocyte pyroptosis could originate from SARS-CoV-2-induced hyperactivation of NLRP3 (nucleotide-binding oligomerization domain-like receptor family, pyrin domain containing 3 activation) inflammasome, resulting in stimulation of caspase-1 activity, that leads to activation and release of potent proinflammatory cytokines including interleukin-1β and interleukin-18 [27]. In fact, IL-1β levels can increase during SARS-CoV-2 infection, and pyroptosis is a paradigmatic hyperinflammatory expression of programmed cell death ensuing from the bioactivity of cytopathic viruses [1,28]. SARS-CoV-2-induced lymphopenia is characterized by significantly decreased numbers of memory T helper cells and regulatory T lymphocytes, and these pathologic traits contribute to shape the dysregulated immune response that takes place in severe COVID-19 patients [29].

In addition to a quantitatively defective lymphocytic reaction, in severely ill COVID-19 patients, the immunopathology of SARS-CoV-2 infection may also be due to relevant deviations from the physiologic immune responses implicated in viral clearance. Indeed, a downregulation of interferon biosynthesis has been noticed in severe infections caused by SARS coronavirus [30]. In particular, the immunopathological framework of the worst COVID-19 situations might be characterized by the development of aberrant T cell immunophenotypes, associated with a dysregulated secretory profile of proinflammatory cytokines and chemokines [2]. In this regard, it has been reported that SARS-CoV-2 can trigger a CD4^+^ T lymphocyte commitment towards a pathogenic Th1 lineage, highlighted by an overproduction of IL-6 and GM-CSF [31]. These cytokines favor the activation of CD14^+^ CD16^+^ monocytes, that further secrete IL-6 and can move from blood to lung, thereby eventually differentiating into alveolar macrophages or dendritic cells [31]. Moreover, severe COVID-19 illness can be marked by dysfunctional CD4^+^ and CD8^+^ T cell immunophenotypes, featured by a concomitant expression of surface markers including PD-1 (programmed cell death protein-1) and Tim-3 (T-cell immunoglobulin and mucin-domain containing-3), associated with an increased susceptibility to the functional exhaustion of T lymphocytes during the course of viral infections [32,33,34,35,36].

Immunoactivation impairment due to functional exhaustion and unresponsiveness of T cells is at least in part attributable to the negative impact exerted on immune defences by cellular senescence, which makes older patients more exposed than young subjects to the risk of developing the most severe pulmonary and systemic COVID-19 complications [37]. The latter often arise from a combination of insufficient adaptive immune responses and hyperactive innate immune pathways, leading to excessive inflammation and tissue damage [36].

It is also noteworthy that in critically ill COVID-19 patients, SARS-CoV-2 can drive the adaptive immune response towards a predominance of the Th17 lymphocytic pattern, leading to activation of chemotactic/proinflammatory circuits responsible for neutrophilic inflammation [37]. Indeed, high levels of interleukin-17 (IL-17) were found in COVID-19 patients with pneumonia [37]. In these subjects, an immunological shift towards hyperactivation of the Th17/IL-17 axis can be favored by several comorbidities including hypertension, obesity, diabetes, and chronic kidney disease [38]. It is thus possible that these pathological conditions predispose some subjects to develop an aberrant upregulation of IL-17A production during SARS-CoV-2 infection, which in turn strengthens neutrophilic inflammation and contributes to dampen the antiviral adaptive immune responses [38].

### 2.3. Role of IL-6 in SARS-CoV-2-Induced Cytokine Storm

The immunopathology of the SARS-CoV-2-induced cytokine storm is quite complex, and not yet fully elucidated. A currently accredited hypothesis suggests that in severely ill COVID-19 patients, the cytokine storm develops as a consequence of the inability of the immune system to clear SARS-CoV-2 [39]. Namely, lymphopenia and the associated defective production of antiviral type I and III interferons would trigger an exuberant inflammatory reaction sustained by excessive innate immune mechanisms, operated by monocytes/macrophages and lung epithelial cells [40]. This hyperinflammatory response is characterized by a very intense release of cytokines and chemokines such as interleukins-2 (IL-2), 7 (IL-7), 8 (IL-8), 18 (IL-18), 33 (IL-33), GM-CSF, IP-10, monocyte chemoattractant protein-1 (MCP-1), macrophage inflammatory protein-1α (MIP-1α), and especially IL-1β, TNF-α, and IL-6 (Figure 1) [2,40,41]. The latter is a pleiotropic cytokine whose serum levels are positively correlated with COVID-19 severity and also appear to be enhanced in association with the observed increase in the numbers of proinflammatory CD14^+^ CD16^+^ monocytes [31,42].

In comparison to influenza and parainfluenza viruses, coronaviruses are known to be capable of eliciting the release of larger quantities of IL-6 from human epithelial cells [43]. IL-6 exerts its biological actions on several cellular targets (Figure 2). At the level of the immune system, IL-6 stimulates the development of Tfh cells, and together with TGF-β contributes to the differentiation of Th17 lymphocytes [7,44]. Moreover, IL-6 inhibits Th1 cell-dependent antiviral responses and impairs the functions of CD8^+^ cytotoxic and natural killer T cells [7,12]. Indeed, IL-6 overexpression seems to be associated with low numbers of CD4^+^ and CD8^+^ T lymphocytes [35].

## 3. Molecular Mechanisms Underlying IL-6 Bioactivities

At the level of target cells, IL-6 binds to its receptor (IL-6R), which is an 80 kDa membrane protein lacking signaling properties [45,46]. Hence, the IL-6/IL-6R molecular complex interacts with another membrane protein, named glycoprotein 130 (gp130), which dimerizes and activates intracellular signaling pathways [47]. This mechanism is known as classic cis-signaling (Figure 3) [48,49]. IL-6R is present only in a few cell types, including leukocytes, epithelial cells, and hepatocytes, whereas gp130 is ubiquitously expressed [50]. IL-6 specifically binds to IL-6R, but cannot interact with gp130 by itself, so that cells harboring gp130 but not IL-6R are insensitive to IL-6 [46]. On the other hand, gp130 is not selective for the IL-6/IL-6R complex, but acts as a signaling module for several other cytokines of the IL-6 family, including interleukin-11 (IL-11), leukemia inhibitory factor, ciliary neurotrophic factor, oncostatin, cardiotrophin-1, and cardiotrophin-like cytokine factor 1 [51]. Cis-signaling mediates multiple biological actions of IL-6 on both adaptive and innate arms of the immune system, thereby affecting the functions of T and B cells, natural killer cells, neutrophils, and macrophages, as well as contributing to the pathogenesis of cytokine storm [49].

In addition to functioning as a membrane-bound receptor, IL-6R can also exist as a soluble form (sIL-6R), originating from the proteolysis catalyzed by the protease “a disintegrin and metalloproteinase domain-containing protein 17” (ADAM17) [52,53]. Differently from other soluble receptors which behave as competitive antagonists thus inhibiting ligand binding to cell membrane receptors, IL-6/sIL-6R rather interacts with gp130, thereby triggering its dimerization and the subsequent intracellular signal transduction (Figure 3) [46]. This mechanism, which has been defined IL-6 trans-signaling [49], makes it possible for cells devoid of IL-6R to respond to IL-6. IL-6 trans-signaling can play a key role in systemic cytokine storm by stimulating the production of IL-6 itself, MCP-1, IL-8, and vascular endothelial growth factor (VEGF), as well as by upregulating the expression of the adhesion molecule E-cadherin [49]. VEGF and E-cadherin induce a marked increase in vascular permeability and leakage, thus significantly contributing to lung damage [49].

Moreover, a further modality of IL-6 signaling refers to the so-called IL-6 trans-presentation (Figure 3) [54]. Namely, this third mechanism is mediated by dendritic cells expressing IL-6R, which present the plasma membrane-bound IL-6/IL-6R complex to antigen-specific T lymphocytes, that in turn contribute to implementing IL-6 signaling via their gp130 [54]. IL-6 trans-presentation drives T cell commitment towards a highly tissue-destructive immunophenotype [46,54].

The above three IL-6 signaling modules converge on dimerization of gp130, whose cytoplasmic transduction subunit triggers the activation of the associated Janus kinase 1 (JAK1) [47,55]. JAK1 phosphorylates some intracellular tyrosine residues of gp130 and induces the phosphorylation-dependent activation of signal transducers and activators of transcription 1 (STAT1) and 3 (STAT3) [46]. Upon JAK1-operated activation, STAT1 and STAT3 homodimerize or heterodimerize and migrate to the nucleus, where they act as transcription factors that promote the activity of gp130 target genes (BIRC5, BCL2, cyclins, NOTCH1, MYC), implicated in cell growth and proliferation [46]. Via this intricate transduction network, IL-6 also activates the suppressor of the cytokine signaling 3 (SOCS3) pathway, thus inhibiting STAT4 phosphorylation and suppressing the functions of CD8^+^ cytotoxic T lymphocytes and natural killer cells [2]. In addition, JAK1-catalyzed phosphorylation of gp130 is also involved in the activation of the signaling units which sequentially include the RAS–mitogen-activated protein kinase (MAPK) cascade and the phosphoinositide 3-kinase (PI3K)/Akt module [46].

Through all the above signaling mechanisms, IL-6 exerts its pleiotropic functions on both immune/inflammatory and structural cells, thus significantly contributing to lung damage via induction of tissue infiltration sustained by neutrophils and macrophages, as well as by promoting vascular permeability, alveolar edema, and hypoxia [2,56]. Moreover, IL-6 can also activate TGF-β-dependent signaling pathways in lung fibroblasts, thereby favoring pulmonary fibrosis and the consequent further impairment of gas exchange [57,58].

## 4. Tocilizumab: Mechanism of Action and COVID-19 Clinical Trials

Anti-IL-6 antibodies inhibit classic cis- and trans-signaling, but do not affect IL-6 trans-presentation, whereas antibodies directed against IL-6R are able to effectively block all three modalities of IL-6 signaling [46]. With respect to IL-6 inhibitors, IL-6R antagonists appear to be safer and therapeutically more effective [46,49]. Among IL-6R blockers, the recombinant humanized monoclonal IgG1κ antibody tocilizumab exerts its pharmacological effects by interacting with the IL-6 binding epitope of IL-6R, thereby preventing IL-6 attachment and inhibiting cis-signaling, trans-signaling, and trans-presentation [54,59,60]. Because of its anti-inflammatory properties, tocilizumab has been approved for treatment of rheumatoid arthritis and systemic juvenile idiopathic arthritis [14,49,61].

The molecular structure of tocilizumab consists of two heavy and two light chains, and includes 12 intrachain and 4 interchain disulphide bonds (Figure 4), globally amounting to a molecular weight of 149 kDa [62]. Tocilizumab is characterized by a nonlinear pharmacokinetic profile, and this drug has a relatively long half-life, approximately ranging from 5 to 12 days [62,63]. In patients with rheumatoid arthritis, tocilizumab is commonly used intravenously at dosages of either 4 mg/kg or 8 mg/kg every four weeks [62]. However, tocilizumab can also be used via the subcutaneous route at a dosage of 162 mg, once every week [62]. Several randomized controlled trials have shown that in patients with rheumatoid arthritis, tocilizumab significantly and persistently improved structural joint damage and health-related quality of life, and also decreased the levels of C-reactive protein [62,63]. Moreover, tocilizumab has been used as an adjuvant therapy for the uncontrolled inflammatory state which characterizes hemophagocytic lymphohistiocytosis associated with visceral leishmaniasis [64]. The most frequently reported side effects and adverse reactions include infections, nausea, mouth ulcerations, abdominal pain, gastritis, hypertension, and headache. With regard to drug–drug interactions, tocilizumab inhibits the downregulation of the cytochrome P450 enzyme (CYP) system induced by IL-6 [62]. Therefore, these pharmacologic aspects should be carefully taken into consideration when tocilizumab is used concomitantly with other drugs metabolized by the CYP enzymatic complex such as warfarin, cyclosporine, theophylline, and phenytoin [62].

### 4.1. Clinical Studies Supporting the Use of Tocilizumab in COVID-19 Patients

Some studies suggest that tocilizumab can be effective in COVID-19 patients [49]. In particular, preliminary data obtained by Xu et al. in a small group (21 subjects) of severe COVID-19 patients evidenced that tocilizumab suppressed fever, lowered C-reactive protein, increased blood lymphocyte count, decreased oxygen intake, and improved computed tomography (CT) scan imaging [65]. According to Perrone et al., authors of the single-arm multicenter TOCIVID-19 prospective trial, designed as a phase 2 study and enrolling about 300 COVID-19 patients, tocilizumab decreased the mortality rate at 30 days, without causing significant toxicity [66]. An observational investigation carried out by Price et al. in 239 COVID-19 patients indicated that tocilizumab prolonged overall survival, improved oxygenation, and decreased the levels of inflammatory biomarkers [67]. A single-center study was conducted on 100 consecutive COVID-19 patients by Toniati et al., who reported that tocilizumab induced a quick and prolonged clinical improvement [68]. A retrospective cohort study, performed by Eimer et al., suggested that tocilizumab shortened the duration of hospitalization and reduced the need for mechanical ventilation and admission to intensive care unit (ICU) [69]. A further retrospective case-control study, carried out by Potere et al. on consecutive COVID-19 patients with pneumonia, hyperinflammation, and hypoxemia, demonstrated that in comparison to standard of care, add-on therapy with tocilizumab reduced mortality and decreased the risk of progression from severe to critical disease [70]. Moreover, a systematic review and meta-analysis performed by Berardicurti et al., including 22 studies and 1520 people treated with tocilizumab, showed that subjects undergoing therapy with this drug were characterized by a lower mortality, when compared to the overall mortality observed in severe COVID-19 patients [71]. With regard to the analysis of 3924 COVID-19 patients admitted to ICU, and recruited among 4485 adults referring to 68 US hospitals, Gupta et al. demonstrated that the death risk resulted in being lower in participants treated with tocilizumab, when compared to subjects not undergoing drug therapy with this anti-IL-6R antagonist [72].

### 4.2. Clinical Studies Questioning the Use of Tocilizumab in COVID-19 Patients

However, after randomly assigning hospitalized patients with COVID-19 and moderate or severe pneumonia to add-on treatment with tocilizumab (64 subjects) or usual care alone (67 subjects), Hermine et al. found that anti-IL-6R therapy lowered the need for mechanical ventilation and the death risk after 14 days, but did not change mortality on day 28 [73]. A retrospective study made by Okoh et al. in 77 COVID-19 patients showed that early treatment with tocilizumab improved the oxygenation pattern during hospitalization, but did not modify the overall survival [16]. Another retrospective investigation, conducted by Campochiaro et al. on 65 severely ill COVID-19 patients with hyperinflammatory status, indicated that after 28 days of treatment, tocilizumab did not induce any improvement in clinical outcomes and mortality rate, when compared to standard therapy [74]. A recent systematic review published by Cortegiani et al., based on 28 clinical studies, suggests that current evidence regarding the efficacy and safety of tocilizumab for COVID-19 treatment is largely observational and methodologically not reliable [75]. These considerations corroborate the indications inferred from a previous systematic review and meta-analysis carried out by Lan et al., who examined 7 retrospective studies referring to 592 adults with severe COVID-19, subdivided in 240 subjects belonging to the tocilizumab group, and 352 assigned to the control arm [15]. All-cause mortality was slightly lower in the tocilizumab group (16.3%), when compared to the control arm (24.1%). However, such a difference did not reach the threshold of statistical significance [15]. Furthermore, the requirement for mechanical ventilation and the risk of ICU admission were similar in the two groups. Therefore, the authors concluded that currently available evidence does not support the therapeutic use of tocilizumab in COVID-19, because this drug does not seem to provide any additional benefit with regard to clinical outcomes [15].

### 4.3. Most Recent Studies Evaluating the Use of Tocilizumab in COVID-19 Patients

In four recent publications in the New England Journal of Medicine, discordant results were reported and further doubts were cast about the therapeutic efficacy of tocilizumab in COVID-19 patients. In the first one, Stone et al. enrolled in a randomized, placebo-controlled trial 243 hospitalized patients with SARS-CoV-2 infection and pulmonary hyperinflammation [76]. After 14 days of treatment, tocilizumab was not able to prevent either intubation or death in such moderately ill subjects [76]. The second study also enrolled hospitalized patients with COVID-19 pneumonia, in which the same outcomes were investigated by Salama et al. in 249 participants treated with tocilizumab, as well as in 128 subjects randomly assigned to the placebo arm [77]. After 28 days of therapy, tocilizumab decreased the probability of progression towards mechanical ventilation, but did not prolong the overall survival [77]. The third study refers to the Randomized, Embedded, Multifactorial Adaptive Platform Trial for Community-Acquired Pneumonia (REMAP-CAP), performed in severely ill COVID-19 patients by Gordon et al., who randomly assigned 353 patients to tocilizumab, 48 subjects to another IL-6 receptor antagonist (sarilumab), and 402 participants to control treatment not including immunomodulators [78]. In comparison to the control arm, both tocilizumab and sarilumab increased the number of organ support-free days, as well as prolonged 90-day survival [78]. However, in the fourth trial (COVACTA study) carried out in patients with severe COVID-19 pneumonia, Rosas et al. did not find any difference with regard to the outcomes analyzed in 294 subjects undergoing treatment with tocilizumab and in 144 individuals randomized to receive placebo [79]. In particular, when compared to placebo, tocilizumab did not improve clinical conditions and 28-day mortality [79].

The main clinical trials evaluating the effects of tocilizumab in the treatment of critically ill COVID-19 patients are summarized in Table 1.

Despite some reports indicating a satisfactory profile of safety and tolerability, the use of tocilizumab has been associated with the possible occurrence of arterial hypertension, leukopenia, severe infections, thrombocytopenia, liver damage, gastrointestinal perforation, skin reactions, and anaphylaxis [80]. Moreover, it has also been suggested that tocilizumab could cause osteonecrosis of the jaws [81].

Finally, it is noteworthy that in seriously ill COVID-19 patients, relevant therapeutic benefits can be obtained by associating tocilizumab in combined treatments with corticosteroids, especially in regard to prevention of all-cause in-hospital mortality [82]. Indeed, the inhibitory effect of tocilizumab on SARS-CoV-2-induced cytokine storm might be potentiated by corticosteroids, which are known to be capable of suppressing the production of many proinflammatory cytokines and chemokines [83].

## 5. Conclusions

There is no doubt that IL-6 plays a pivotal role in the development and amplification of the aggressive cytokine storm underlying the pathobiology of pulmonary and systemic injuries caused by SARS-CoV-2. Therefore, a robust pathophysiologic and pharmacologic rationale supports the use of therapeutic strategies targeting IL-6 or its receptor. Within this context, the antagonistic blockade of IL-6R implemented by tocilizumab appears as a very interesting approach for treatment of severe COVID-19 patients. Indeed, in less than one year, a relatively high number of different types of trials have evaluated the effects of tocilizumab. So far, the growing body of studies assessing the efficacy and safety of tocilizumab for treatment of SARS-CoV-2-associated hyperinflammation, severe pneumonia, and ARDS has yielded mixed results. Promising data suggesting a positive effect of tocilizumab on clinical outcomes, lung failure, risk of intubation, and even mortality are currently contradicted by other reports indicating that anti-IL-6R treatment is unable to affect the above parameters. Such discrepancies may arise from heterogeneous individual responses, which likely depend on the specific pathogenic weight of IL-6 in the complex scenario of the SARS-CoV-2-elicited cytokine storm. It can thus be argued that when IL-6 exerts a dominant function within the immune/inflammatory mechanisms underpinning severe COVID-19, tocilizumab can be quite effective. On the contrary, we can speculate that when the role of IL-6 is overwhelmed by a redundant array of multiple cytokines and chemokines, the potential therapeutic activity of tocilizumab might be significantly inhibited and neutralized. Of course, other potential causes of the above mentioned discrepant data could include both patient recruitment criteria and timing of treatment [84]. Therefore, further and larger controlled clinical trials are required, which should be aimed at understanding if suitable and reliable biomarkers can be identified in order to predict the sensitivity to tocilizumab, as well as trying a tentative personalized approach for a cytokine-targeted treatment of severe COVID-19 patients.

## Figures and Tables

**Figure 1 ijms-22-03059-f001:**
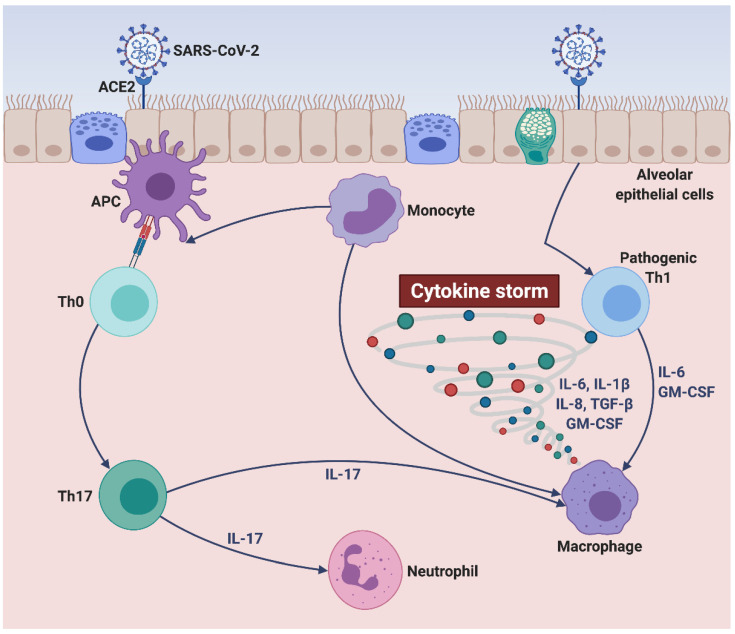
SARS-CoV-2-induced cytokine storm. SARS-CoV-2 targets alveolar epithelial cells, binds to their ACE2 receptors, and penetrates inside lung tissue, thus reaching subepithelial antigen-presenting cells. The latter can drive the differentiation of unpolarized naïve T cells into mature Th17 lymphocytes producing IL-17, leading to neutrophil recruitment and macrophage activation. Moreover, upon their interactions with SARS-CoV-2, infected alveolar epithelial cells can trigger the activation of pathogenic Th1 cells and monocytes/macrophages, which release large quantities of proinflammatory cytokines (IL-6, GM-CSF, IL-1β, TGF-β) and chemokines (IL-8). Within such a cytokine/chemokine milieu (cytokine storm), monocytes can undergo further cellular differentiation towards either alveolar macrophage phenotype or antigen-presenting cell lineage. This original figure was created by the authors using BioRender.com. SARS-CoV-2: severe acute respiratory syndrome coronavirus-2; ACE2: angiotensin-converting enzyme 2; APC: antigen-presenting cell; Th0: unpolarized naïve T helper cell; Th1: T helper 1 cell; Th17: T helper 17 cell; IL-1β: interleukin-1β; IL-6: interleukin-6; IL-8: interleukin-8; IL-17: interleukin-17; GM-CSF: granulocyte-macrophage colony-stimulating factor; TGF-β: transforming growth factor-β.

**Figure 2 ijms-22-03059-f002:**
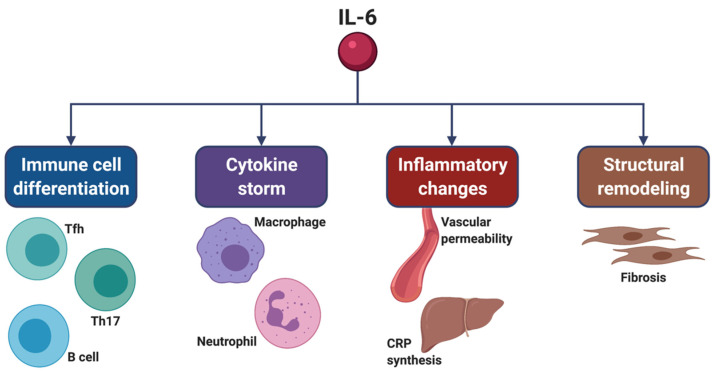
Pleiotropic actions of IL-6, which targets many cell types and tissue districts. IL-6 promotes the differentiation of B cells, as well as of Th17 cells and Tfh cells. By contributing to the differentiation of Th17 cells, IL-6 also induces neutrophil recruitment and macrophage activation, which occur as relevant consequences of cytokine storm. Moreover, IL-6 increases vascular permeability and the hepatic synthesis of CRP, thus significantly contributing to the development and persistence of inflammation. IL-6 also induces tissue remodeling via stimulation of both fibroblast proliferation and production of extracellular matrix proteins. This original figure was created by the authors using BioRender.com. IL-6: interleukin-6; Th17: T helper 17 cell; Tfh: T follicular helper cells; CRP: C-reactive protein.

**Figure 3 ijms-22-03059-f003:**
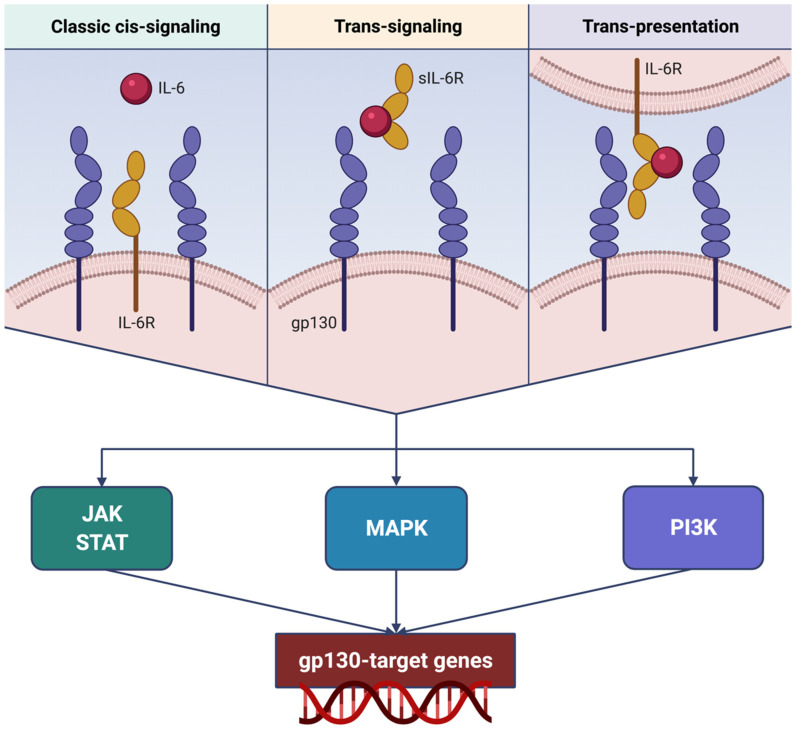
Different mechanisms underlying IL-6R activation and signaling. IL-6 can either interact with membrane-bound IL-6R (classic cis-signaling), or bind to sIL-6R (trans-signaling), or even be trans-presented from dendritic cells through their surface IL-6R to T lymphocytes (trans-presentation). Classic cis-signaling, trans-signaling, and trans-presentation converge on downstream transduction pathways, consisting of dimeric gp130-dependent activation of complex networks including JAK/STAT, MAPK, and PI3K signaling modules. Via these intracellular enzyme systems, IL-6R-triggered biological signals reach the nucleus and stimulate gp130 target genes involved in cell growth and proliferation. This original figure was created by the authors using BioRender.com. IL-6: interleukin-6; IL-6R: interleukin-6 receptor; sIL-6R: soluble interleukin-6 receptor; gp130: glycoprotein 130; JAK: Janus kinases; STAT: signal transducers and activators of transcription; MAPK: mitogen-activated protein kinases; PI3K: phosphoinositide 3-kinase.

**Figure 4 ijms-22-03059-f004:**
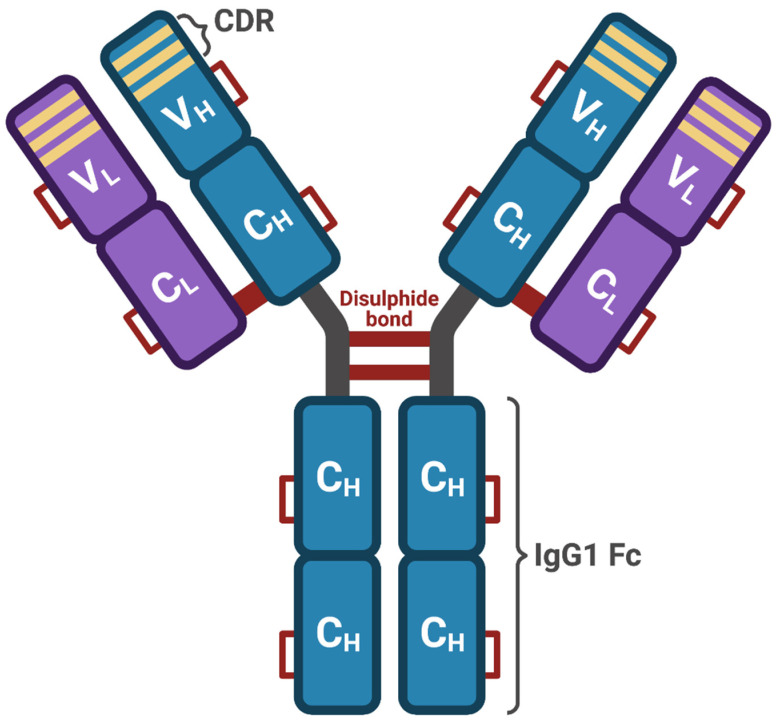
Tocilizumab structure. Tocilizumab is a recombinant humanized IgG1κ antibody, consisting of two identical light chains, each one including a variable (V_L_) and a constant (C_L_) domain, coupled to two identical heavy chains, each one including a variable domain (V_H_) and three constant domains (C_H_). Overall, this antibody structure includes 12 intrachain and 4 interchain disulfide bonds. The variable portion of tocilizumab includes the complementarity-determining regions (CDR), which bind the IL-6 receptor. This original figure was created by the authors using BioRender.com.

**Table 1 ijms-22-03059-t001:** Summary of the main studies evaluating tocilizumab for treatment of critically ill COVID-19 patients.

Authors	Dosage	No. Pt.	Main Results
Xu et al. [66]	4–8 mg/kg	21	Improvements of clinical, laboratory, and radiologic parameters
Perrone et al. [TOCIVID-19] [67]	8 mg/kg	301	Decrease of 30-day mortality
Price et al. [68]	8 mg/kg	239	Decrease of inflammatory biomarkers and survival prolongation
Toniati et al. [69]	8 mg/kg	100	Quick and prolonged clinical improvement
Eimer et al. [70]	8 mg/kg	29	Shortening of hospital stay
Potere et al. [71]	324 mg	/	Decrease of mortality
Gupta et al. [73]	8 mg/kg	3924	Decrease of death risk
Hermine et al. [74]	8 mg/kg	64	No change in 28-day mortality
Okoh et al. [75]	8 mg/kg	77	No change in overall survival
Campochiaro et al. [76]	400 mg	65	No change in 28-day mortality
Stone et al. [79]	8 mg/kg	243	No change in intubation risk and day 14 mortality
Salama et al. [80]	8 mg/kg	389	No change in overall survival
Gordon et al. [REMAP-CAP] [81]	8 mg/kg	353	Prolongation of 90-day survival
Rosas et al. [COVACTA] [82]	8 mg/kg	294	No change in day 28 mortality

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
