# Peer review of "Therapeutic Role of Tocilizumab in SARS-CoV-2-Induced Cytokine Storm: Rationale and Current Evidence"

_ijms, 2021, doi:10.3390/ijms22063059_

Round 1

Reviewer 1 Report

The paper entitled  “Therapeutic role of tocilizumab in SARS-CoV-2-induced cytokine storm: rationale and current evidence" by Pelaia et al. submitted to IJMS addresses the extremely current issues related to the cytokine storm in COVID-19 and current attempts to prevent it. My main doubts are caused by the lack of reference by the authors to numerous data indicating the recently demonstrated benefits resulting from co-treatment with toclizumab and corticosteroids (DEX) in patients with COVID-19. In my opinion, this issue should also be covered in the manuscript. Please try to relate the combined therapy to the effects on the cytokine storm.

Please include also the following minor points in your revision:

  1. Introduction 1: Please Although the Authors focus on discussing COVID-19 in the context of inflammation and cytokine storm in the lungs, please mention in the phenotypic traits of this infection other symptoms of the disease, for example neurological.
  2. Figure 1 should be reorganized in view of the involvement of IL-6 in the cytokine storm. The addition of an overall figure illustrating the pathomechanism of the cytokine storm in COVID-19 may be considered, mainly because the authors refer to it.
  3. Reports indicating the benefits of using toclizumab in combination therapy in patients with COVID-19 should be mentioned.

Author Response

To the attention of:

Reviewer 1

Dear Reviewer 1,

we would like to thank you very much for having carefully reviewed our manuscript, thus suggesting those changes which have significantly improved its overall quality. We have prepared a revised version of this paper, taking into account your comments. The changes have been highlighted in yellow.

The paper entitled  “Therapeutic role of tocilizumab in SARS-CoV-2-induced cytokine storm: rationale and current evidence" by Pelaia et al. submitted to IJMS addresses the extremely current issues related to the cytokine storm in COVID-19 and current attempts to prevent it. My main doubts are caused by the lack of reference by the authors to numerous data indicating the recently demonstrated benefits resulting from co-treatment with tocilizumab and corticosteroids (DEX) in patients with COVID-19. In my opinion, this issue should also be covered in the manuscript. Please try to relate the combined therapy to the effects on the cytokine storm.

Response. In the revised manuscript, we covered the issue regarding the main positive results of the co-treatment tocilizumab-corticosteroids, and we related the outcomes of such a combined therapy to the inhibitory effects on cytokine storm (page 9, lines 361-366).

Please include also the following minor points in your revision:

  1. Introduction 1: Please Although the Authors focus on discussing COVID-19 in the context of inflammation and cytokine storm in the lungs, please mention in the phenotypic traits of this infection other symptoms of the disease, for example neurological.
  2. Figure 1 should be reorganized in view of the involvement of IL-6 in the cytokine storm. The addition of an overall figure illustrating the pathomechanism of the cytokine storm in COVID-19 may be considered, mainly because the authors refer to it.
  3. Reports indicating the benefits of using tocilizumab in combination therapy in patients with COVID-19 should be mentioned.

Response.

  1. With regard to phenotypic traits, neurological symptoms have been mentioned in the revised text (page 2, lines 47-48).
  2. Figure 1 (Figure 2 in the revised manuscript) has been reorganized in the revised manuscript, in view of the involvement of IL-6 in the cytokine storm (page 6). Moreover, another figure has been drawn (Figure 1 in the revised manuscript), illustrating the main pathobiological aspects regarding the cytokine storm in COVID-19 (page 3).
  3. In the revised manuscript, we mentioned the benefits related to the use of the co-treatment tocilizumab-corticosteroids (page 9, lines 361-366).

Sincerely yours,

on behalf of all the authors who contributed to revise the manuscript and approved the final version,

Girolamo Pelaia, MD

Department of Health Sciences

University “Magna Graecia” of Catanzaro

Catanzaro, Italy

Tel.  + 39 0961 3647171

E-mail: pelaia@unicz.it

Reviewer 2 Report

The manuscript submitted to IJMS entitled “Therapeutic role of tocilizumab in SARS-CoV-2-induced cytokine storm: rationale and current evidence” is an original review which aim to focus on the rationale of tocilizumab utilization in SARS-CoV-2-triggered cytokine storm.  On my opinion the article is interesting, well written, with good English.  I highlighted some issues:      minor spell check required     summary of abbreviations required     Introduction: This section has been properly prepared.     Section 2: This section has been properly prepared.     Section 3: This section has been properly prepared.     Section 4: This section could be improved with an update on complication related with tocilizumab administration ( DOI: 10.1016/j.oraloncology.2020.104659).     Conclusions: This section has been properly prepared.       After making the indicated changes, the article may be suitable for publication.

Author Response

To the attention of:

Reviewer 2

Dear Reviewer 2, 

we would like to thank you very much for having carefully reviewed our manuscript, thus suggesting those changes which have significantly improved its overall quality. We have prepared a revised version of this paper, taking into account your comments. The changes have been highlighted in yellow.

The manuscript submitted to IJMS entitled “Therapeutic role of tocilizumab in SARS-CoV-2-induced cytokine storm: rationale and current evidence” is an original review which aim to focus on the rationale of tocilizumab utilization in SARS-CoV-2-triggered cytokine storm.  On my opinion the article is interesting, well written, with good English.  I highlighted some issues:      minor spell check required     summary of abbreviations required     Introduction: This section has been properly prepared.     Section 2: This section has been properly prepared.     Section 3: This section has been properly prepared.     Section 4: This section could be improved with an update on complication related with tocilizumab administration (DOI: 10.1016/j.oraloncology.2020.104659).     Conclusions: This section has been properly prepared.       After making the indicated changes, the article may be suitable for publication.

Response. In the revised manuscript, spelling has been carefully checked throughout the text. Moreover, a summary of abbreviations has been included (page 1, lines 24-41). Furthermore, in the revised Section 4 the potential occurrence of osteonecrosis of the jaws (as a consequence of tocilizumab treatment) has been mentioned (page 9, lines 359-360), according to the reference authored by Bennardo et al. (Oral Oncol 2020;106:104659).

Sincerely yours,

on behalf of all the authors who contributed to revise the manuscript and approved the final version,

Girolamo Pelaia, MD

Department of Health Sciences

University “Magna Graecia” of Catanzaro

Catanzaro, Italy

Tel.  + 39 0961 3647171

E-mail: pelaia@unicz.it